# Deep-Learning-Based Scalp Image Analysis Using Limited Data

**Minjeong Kim** [1,†], **Yujung Gil** [1,†], **Yuyeon Kim** [1] **and Jihie Kim** [2,*]

1 Department of Computer Science and Engineering, Dongguk University, Seoul 04620, Republic of Korea
2 School of Artificial Intelligence Convergence, Dongguk University, Seoul 04620, Republic of Korea
* Correspondence: jihie.kim@dongguk.edu
† These authors contributed equally to this work.

**Abstract:** The World Health Organization and Korea National Health Insurance assert that the number of alopecia patients is increasing every year, and approximately 70 percent of adults suffer from scalp problems. Although alopecia is a genetic problem, it is difficult to diagnose at an early stage. Although deep-learning-based approaches have been effective for medical image analyses, it is challenging to generate deep learning models for alopecia detection and analysis because creating an alopecia image dataset is challenging. In this paper, we present an approach for generating a model specialized for alopecia analysis that achieves high accuracy by applying data preprocessing, data augmentation, and an ensemble of deep learning models that have been effective for medical image analyses. We use an alopecia image dataset containing 526 good, 13,156 mild, 3742 moderate, and 825 severe alopecia images. The dataset was further augmented by applying normalization, geometry-based augmentation (rotate, vertical flip, horizontal flip, crop, and affine transformation), and PCA augmentation. We compare the performance of a single deep learning model using ResNet, ResNeXt, DenseNet, XceptionNet, and ensembles of these models. The best result was achieved when DenseNet, XceptionNet, and ResNet were combined to achieve an accuracy of 95.75 and an F1 score of 87.05.

**Keywords:** ensemble; data augmentation; alopecia

## 1. Introduction

According to Korea National Health Insurance [1] in 2021, the number of alopecia patients in Korea increased from 103,000 in 2001 to 145,000 in 2005 and then surged to 233,000 in 2020. As these figures do not reflect patients whose alopecia is caused by genetic factors and aging, the domestic alopecia population is estimated to be approximately 10 million as of 2021. Early treatment is known to be effective for alopecia because the symptoms worsen over time [2]. Therefore, it is essential to detect alopecia early. Currently, to diagnose and prevent alopecia, people need to personally visit a specialized scalp clinic. Owing to the increase in the population of alopecia sufferers and consequent interest in preventing alopecia, personal consultations with experts are likely to involve a considerable amount of delay. People can also feel burdened just by visiting a clinic because they spend time and money in clinics. Accordingly, there is a need for a convenient and accessible solution to diagnose and analyze alopecia. Therefore, systems that allow users to easily detect alopecia at home are being actively studied. Several systems have attempted to diagnose alopecia by extracting alopecia characteristics [3] and analyzing the thickness and density of the hair [4], and by using microscope images captured under a portable camera that can be detached from a smartphone. Most of these systems use general image data processing approaches rather than specific data processing customized for analyzing alopecia.

Our contributions are as follows: (1) we created an alopecia-specialized model that can achieve high accuracy even with a limited dataset; (2) we present several data augmentation approaches that are appropriate for alopecia images; (3) we evaluate the performance of ResNet, ResNeXt, DenseNet, and XceptionNet that have been effective for medical data

analyses, and then ensemble the models to create alopecia-specialized models. The accuracy was highest when ensembling DenseNet, XceptionNet, and ResNet, specifically, 95.75 (with an F1 score of 87.05).

The rest of this paper is organized as follows: Section 2 reviews existing research on scalp data preprocessing methods and existing models for scalp datasets. Section 3 presents our data preprocessing method, our data augmentation method, and the models that we used for alopecia condition classification. Section 4 includes a description of the model architecture and the results of this research. Section 5 concludes this paper.

## 2. Literature Review

### 2.1. Scalp Data Preprocessing Methods

Kim et al. [3] implemented a method to measure hair density, which is the most basic feature used to diagnose alopecia, through image processing and scalp hair microscope image datasets. As previous research has diagnosed alopecia based on the extent of hair growth, it is necessary to cut the hair. The method by Kim et al. [3] is meaningful in that it uses preprocessing with contrast stretching and morphology processing, converts skeleton images, and applies a search algorithm to identify the endpoint to measure density. However, as this approach focuses only on the density of hair, it has the limitation that it ignores other characteristics of early detection such as dead skin cells and erythema.

Kim et al. [4] implemented an alopecia diagnosis system using hair density, thickness, number of hair follicles, and redness, which are indicators that can determine alopecia. Similar to Kim et al. [3], this work uses scalp hair microscope image datasets and a similar preprocessing method. However, the average distance between two points is determined by applying Canny Edge Detection to detect the thickness. The RGB values, especially R values, were compared to determine redness. Although the work considers more diverse alopecia indicators than Kim et al. [3], it does not conduct preprocessing based on the scalp characteristics of each person, and the likelihood of alopecia cannot be expressed in a percentage form.

ESENSEI data mining [5] reduces and normalizes the image to lower the difference in the hue between the images when predicting the location of hair follicles in scalp images. A single image can be used to create eight images by using x and y axis symmetry. As it only uses symmetry, it has the advantage that interpolation is not necessary. However, the degree of enhancement is low.

Trichoscopy of alopecia areata [6] is an algorithm for diagnosing alopecia by extracting HLF from the images captured via a microscope. In Seo et al. [6], image datasets are converted to gray tones to reduce errors arising from color differences. Contrast stretching is used to recover the shadow of the microscope and the area covered by the light reflection of the scalp. Techniques such as contrast stretching reduce noise in images. However, its disadvantage is that it is weak against color changes in hair because it processes images only in gray tones. HLF consists of a hair count, thickness estimation, and follicle count, and each HLF is trained in Seo et al. [6]. It is suitable for measuring the features by dividing them as per each cause of alopecia, but it is difficult to use it to consider scalp diseases except the hair condition.

Reference [7] flipped the photo vertically and preprocessed it by a $+15-15°$ rotation. As a result, in EfficientDet and DetectirRS, mAP (50:95) increased performance by about five to 20, but in YOLOv4, it increased by about one to two, so the performance improvement was small. This is due to the mosaic enhancement of YOLOv4. Mosaic augmentation synthesizes four training images so that it can learn to detect small objects.

Table 1 summarizes the existing scalp data preprocessing works.

**Table 1.** Summary of the existing scalp data preprocessing works.

| Reference | Description | Problems |
|---|---|---|
| [3] | Diagnosed alopecia based on the extent of hair growth. Preprocessed using feature extraction algorithm. | Focused only on the density of hair. |
| [4] | Used various indicators that can determine alopecia, such as hair density, thickness, and so on. | There was no preprocessing to reflect the characteristics of each person. |
| [5] | Normalized the images to lower the difference in hue between the scalp images. | The degree of enhancement is low. |
| [6] | Converted to gray tones to reduce errors arising from color differences and created an algorithm for diagnosing alopecia by extracting HLF from scalp images. | Difficult to use it to consider scalp diseases, except for the hair condition. |
| [7] | Flip images vertically and rotate +15 to −15° | Small performance improvement on models with data augmentation techniques such as YOLOv4. |

## 2.2. Existing Models for Scalp Datasets

Kim et al. [7] implemented an automated measurement of hair density using deep neural networks. In [7], a model that is simply learns hair follicle images and detects hair follicles within a scalp image. Thereafter, hair loss was determined through the number of hair follicles appearing in the image. Reference [8] describes an automatic trichoscopic image analysis model. This model consists of D-Net for trichoscopy image detection and R-Net for prediction. If a trichoscopic image is used as the input of D-Net, then the hair follicle is detected. When this process is finished, the R-Net calculates the number of hairs as well as the proportion of hairs of different types. In the case of the methods in [7,8], if the shape of the race, the shape of the hair follicle, or the distance that the photograph was taken changes, showing low accuracy. In fact, the maximum accuracy of [7] is 75.73%.

ScalpEye [9] is an intelligent scalp examination and diagnosis system based on deep learning for scalp physical therapy. Images of hair under a microscope are used as a training dataset. The system uses Faster R-CNN with the Inception ResNet_v2_Atrous model for examining scalp hair symptoms. The cost and time of educating and training scalp physical therapists can be reduced. However, there is a limitation in that it does not directly serve the user.

Benhabiles et al. [10] designed a system that uses facial images as a dataset to detect alopecia and classify it into seven levels. To reduce overfitting, this approach applies a horizontal reflection, gaussian noise, gaussian blur, and contrast-limited adaptive histogram equalization for data enhancement. As Benhabiles et al. [10] only use methods for general image enhancement, not scalp-specific enhancement processes, it is inadequate to be applied to scalp-specialized models.

Shakeel et al. [11] proposed a framework to classify healthy hairs and alopecia areas using support vector machine (SVM) and K-nearest neighborhood (K-NN). It uses only 200 healthy hair images and 68 alopecia area images. In order to overcome the disadvantage of the small amount of data, Shakeel et al. [11] use image preprocessing and enhancement through histogram equalization (HE). Three features, i.e., hair color, hair texture, and hair shape are extracted. Although Shakeel et al. [11] is meaningful in that each hair feature is extracted and trained; however, it suffers from a high possibility of overfitting because only HE is used for data enhancement.

Ref. [12] used CNN for image classification by automatically extracting features from raw pixel data. The model included ReLU activation, pooling layers to reduce feature map

dimensions, and dropout layers to prevent overfitting. A 0.3 dropout rate resulted in 30% of the neurons being dropped randomly in each epoch. However, this model can only judge if the image is alopecia or non-alopecia, so it is difficult to diagnose the progress of one's alopecia.

Reference [13] compares the performance of various machine learning algorithms, including SVM, KNN, the Random Forest classifier, Gaussian Naive Bayes, and CNN, to accurately classify them as alopecia symptoms. Overall, [13] aims to improve the accuracy of dermatology alopecia diagnosis using machine learning techniques. The CNN algorithm showed the highest accuracy at 92%. Reference [14] classified scalp lesion images by adding a convolutional block attention module (CBAM) and spinal FC to the DenseNet classic model. In addition, combining cloud computing and AIoT design architecture can be used in more general situations. As a result, an accuracy of 85.03% was obtained. However, the accuracies in [13,14] are insufficient to diagnose scalp diseases such as alopecia and scalp lesions.

Reference [15] generated additional data using data augmentation (DA), and features were extracted using a VGG-19 pretrained CNN model. Reference [15] proposed VGG-SVM is proven to be 98.31% accurate in simulations using 200 HH images from Figaro1k datasets and 68 AA images from Dermnet datasets. They have a limitation in that they did not take into account racial differences.

Table 2 summarizes the existing models for scalp data.

**Table 2.** Summary of the existing models for scalp data.

| Reference | Description | Problems |
|-----------|-------------|----------|
| [7] | A model that is simply learned hair follicle images and detects hair follicles within a scalp image. | If the race, the shape of the hair follicle, and the location or distance of the picture are different, the accuracy is low. |
| [8] | An automatic hair follicle image analysis model which consists of D-Net for trichoscopy image detection, and R-Net for prediction. | |
| [9] | A system that uses Faster R-CNN with the Inception Res-Net_v2_Atrous model for examining scalp hair symptoms. | There is a limitation in that it does not directly serve the user. |
| [10] | A system that uses facial images as a dataset to detect alopecia and classify it into seven levels. | It only uses methods for general image enhancement, and not scalp-specific enhancement processes. |
| [11] | A framework that consists of a support vector machine and a K-nearest neighborhood. | It suffers from a high possibility of overfitting because only HE is used for data enhancement. |
| [12] | An image classification model, which consists of CNN structure, that extracts the characteristics of alopecia automatically. | It can only judge if the image is alopecia or non-alopecia, so it is difficult to diagnose the progress of one's alopecia. |
| [13] | They preprocessed the dataset using image enhancement, segmentation, and data augmentation techniques. They compared the performance of various machine learning algorithms. | The accuracy is insufficient to diagnose scalp diseases, such as alopecia and scalp lesions. |
| [14] | A scalp lesion image classifier that combines cloud computing and AIoT design architecture with an algorithm that adds a convolutional block attention module (CBAM) and spinal FC to the DenseNet classic model. | |

**Table 2.** *Cont.*

| Reference | Description | Problems |
|---|---|---|
| [15] | They proposed VGG-SVM for alopecia diagnosis, and their algorithm showed the highest accuracy at 98.31%. | They did not take into account racial difference. |

## 3. Method Description

### 3.1. Dataset

In this study, an open dataset from AI Hub [16] was used, which was reviewed by three Seoul National University Hospital specialists who set the classification criteria. Data augmentation was performed to increase the amount of data. Scalp images were taken from four sides, i.e., the top of the head, left, right, and larynx, were included, and the training was conducted using these scalp images. In the original dataset, four distinct alopecia conditions were annotated: 526 good (0), 13,156 mild (1), 3742 moderate (2), and 825 severe (3). The numbers 0, 1, 2, and 3 are used to denote each of the conditions. The dataset is rather unbalanced, i.e., there is a high variance in the number of examples for each condition. Example images for each label are presented in Table 3.

**Table 3.** Example image for each label.

| Label | Good (0) | Mild (1) | Moderate (2) | Severe (3) |
|---|---|---|---|---|
| Image | 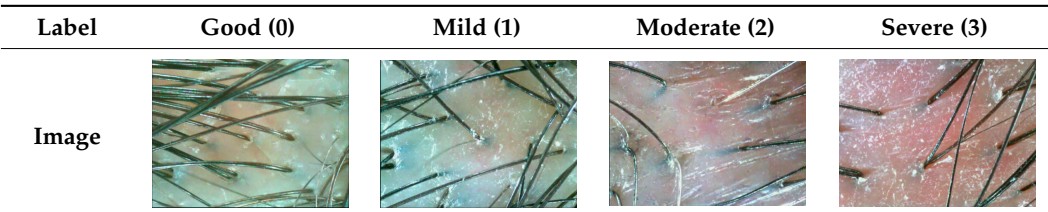 | | | |

In this study, 18,249 original alopecia data were used. To reduce the imbalance of the data, these data were increased by 6–10 times, depending on the distribution of the classes, to 49,118 by applying the data augmentation approaches described in Section 3.2. Then, 39,545 data, 80% of total data, was used as train data, and 9573 data, 20% of all data, as test data. The overall data processing process is shown in Figure 1. When the above process is applied, the augmentation is approximately 6 to 10 times.

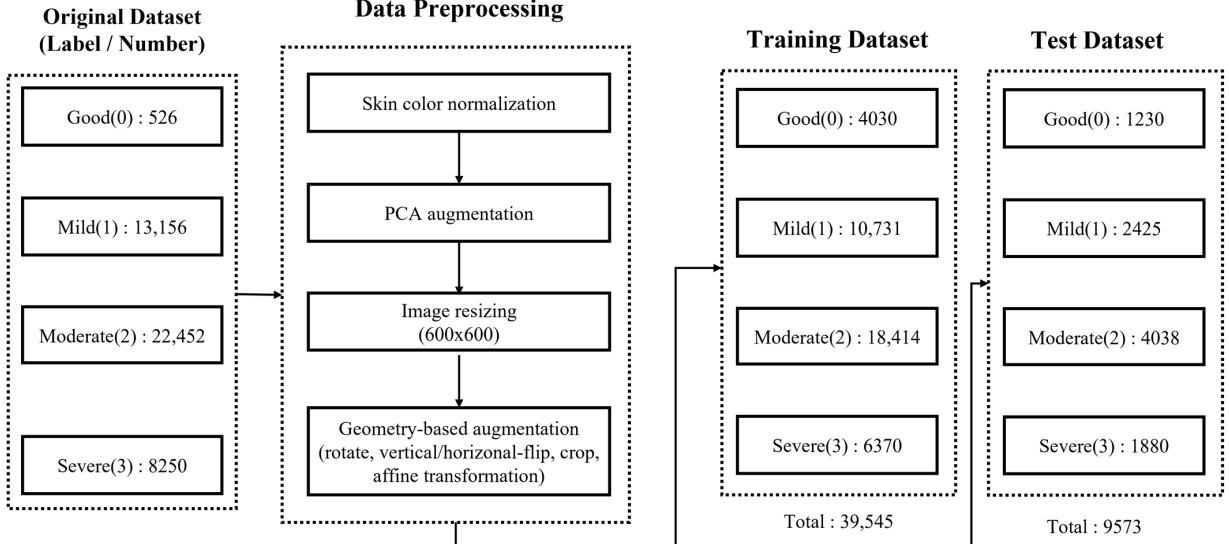

**Figure 1.** Data preprocessing and data augmentation of scalp images.

*3.2. Data Augmentation*

Each original data was photographed in a different environment, and thus it has different size and background. Therefore, some data has a background that is not related to scalp information, and noise, such as a shadow of a camera that exists at the edge of the data. To remove these noises, each image is cropped to a square of 600 pixels based on the center of the data. Since the original data is over 600 pixels, the shadow and noise appearing at the edge are removed by cropping. Shadows that are not removed by cropping are faded through a preprocessing process, such as color conversion, after the current step. In other words, this paper used 600 × 600 resized images as input data. This can also reduce the learning resources that are required for deep learning models.

Jakubik [5] asserted that the accuracy is the best when the augmentation related to image transformation is applied in training as well as test datasets. The dataset of the scalp is insufficient to generate a high-accuracy model; therefore, data augmentation was added to the original datasets. In the case of the scalp images, the data present very similar characteristics when they are flipped up, down, left, and right; therefore, images were turned vertically and horizontally, rotated 90°, and then rotated in additional 10° units. Affine transformation was necessary to distort the scalp images, considering the input from various angles. We distort the color of the datasets in order to increase the accuracy of the images obtained from patients of diverse races and hair colors. The pixel data are then divided by 255 to a scale of (0, 1) intervals for simple normalization so that they can be used as inputs to the CNN-based networks.

The color of the skin varies depending on the lighting. As such, variation is disadvantageous in this analysis, and we apply a normalization process. We extract the color from a reference image and make the color of input images similar to the reference image. In the dataset, it was found that skin color existed within the spectrum of red to blue. We determine one reference image that we think is in the middle range. Then, the color extracted from this image is used to normalize other images. Equations (1)–(3) represent the afore-described process. First, $M$ is calculated by obtaining the average of each of the RGB pixel values of image $I_{xy}$. $L$ is then computed by obtaining the average of all pixels of image $I_{xy}$. The color is extracted from the reference image $I_r$ using the color extraction algorithm, and the color indicators $M_r$ and $L_r$ are calculated. The color indicators $M_i$ and $L_i$ for input images $I_i$ are also computed in the same manner. Then, we modify the input images $I_i$ to create processed images $I_p$ using Equation (4). This calculation is performed by adding a color index difference between the reference image and the input image. The dataset contained many images having red, yellow, white, green, and blue colors. One from each of these was selected as an example image $I_i$ to analyze the results of image processing for each color. Any image may be used as a reference image $I_r$ for extracting a color. We selected the white image as the reference image among the example images. This is because white is expected to reduce the effect of deep neural networks owing to color bias because red, green, and blue colors are the most uniform. Table 4 presents the results of generalizing skin tone for example images by the color classification. Green and blue images were satisfactorily changed, but red and yellow were not. To overcome this issue, we proceeded with the addition of red in the next step.

$$I_{xy} = \left[ \, I_{xy}^R \, , I_{xy}^G \, , I_{xy}^B \, \right] \tag{1}$$

$$M = \left[ M^R, \, M^G, \, M^B \right] = \left[ \frac{\sum I_{xy}^R}{H \times W}, \frac{\sum I_{xy}^G}{H \times W}, \frac{\sum I_{xy}^B}{H \times W} \right] \tag{2}$$

$$L = \left[ \frac{\sum I_{xy}^R + \sum I_{xy}^G + \sum I_{xy}^B}{3 \times H \times W} \right] \tag{3}$$

$$I_p = I_i + (M_r - M_i) + (L_i - L_r) \tag{4}$$

$$I_{rp}^R = I_p^R + \gamma \times M_r^R, \ (\gamma | \gamma \in raito) \tag{5}$$

$$I_{pca} = I_{xy} + [p_1, \ p_2, p_3][\alpha\lambda_1, \ \alpha\lambda_2, \ \alpha\lambda_3] \tag{6}$$

**Table 4.** Examples of skin color normalization.

| | Red Image | Yellow Image | $I_r$ | Green Image | Blue Image |
|---|---|---|---|---|---|
| $I_i$ (input) | | | | | |
| $I_p$ (result) | | | | | |

As a strategy for data augmentation, we generate new images by adding a red tone to the existing images. The red addition is calculated simultaneously with the previously mentioned skin tone normalization. We use *N*(e.g., 5), which indicates how many images will be augmented, and a range *ratio* for how much red we want to add (e.g., [−0.2, 0.2]). A method of calculating the processing image $I_{rp}$ from the skin tone normalized image $I_p = \left[ I_p^R, \ I_p^G, \ I_p^B \right]$ is given in Equation (5). The $M_r^R$ value of the reference image is multiplied by $\gamma$, which is a random value from *ratio*. The *ratio* is a range of arbitrarily determined values. The $\gamma \times M_r^R$ value is added to the red value of the input image $I_p^R$. This process is repeated *N* times. As for the user input parameter, the number of output images *N* was set to 5, and the red processing range *ratio* was set between −20% to 20%. The parameter was carefully determined to be a value such that the red color was well added and removed in all the randomly sampled images. The red tone was randomly added or subtracted so that the existing amount of red was less affected by the deep neural network learning. Table 5 presents an example of red addition and augmentation. $I_p$ denotes the input and $I_{rp}$ denotes the result.

For more color transformation, PCA augmentation described in Krixhevsky et al. [17] was used. First, $\alpha$, which denotes variance, was determined. The process of calculating the PCA augmented image $I_{pca}$ from the input image $I_{xy} = \left[ I_{xy}^R, \ I_{xy}^G, \ I_{xy}^B \right]^T$ is as given in Equation (6), where $p_i$ and $\lambda i$ are the $i^{\text{th}}$ eigenvector and eigenvalue of the 3 × 3 covariance matrix of $I_{xy}$, respectively. The $p_i$ and $\lambda i$ values are calculated by $I_{xy}$. We set the parameter $\alpha$ value to 0.3. The reason is that if the value is higher than 0.5, some pixels become abnormal as shown in Figure 2, and with a value of 0.1, the change in the RGB pixel value is very small, i.e., between −1 to 1. Table 6 presents the results of PCA augmentation; there is no significant difference in sight, but the neural network can make use of the difference.

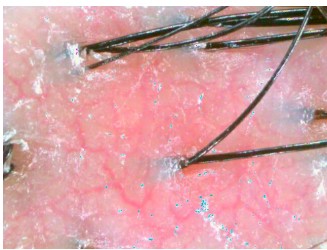

**Figure 2.** Some pixels are overprocessed and show a different color than the surroundings when $\alpha$ has a value higher than 0.5.

**Table 5.** Examples of red addition and augmentation.

| | Red Image | Yellow Image | $I_r$ | Green Image | Blue Image |
|---|---|---|---|---|---|
| $I_p$ |  |  |  |  |  |
| $I_{rp}$ |  |  |  |  |  |

**Table 6.** Examples of PCA Augmentation.

| | Red Image | Yellow Image | White Image | Green Image | Blue Image |
|---|---|---|---|---|---|
| $I_{xy}$ (input) |  |  |  |  |  |
| $I_{pca}$ (result) |  |  |  |  |  |

*3.3. Loss Function*

As described in Section 3.1, the dataset is highly unbalanced. For training the neural network, we adjusted the focal loss function for the unbalanced dataset. The number of training data is calculated for each class (good, mild, moderate, and severe), and each weight used in the CNN-based model is calculated using Equation (7) below.

$$total_{num} = train_{dict[0]} + train_{dict[1]} + \ldots + train_{dict[3]} \tag{7}$$

$$class_{weight} = \left[ 1 - \frac{train_{dict[0]}}{total_{num}}, \ 1 - \frac{train_{dict[1]}}{total_{num}}, \ldots, \ 1 - \frac{train_{dict[3]}}{total_{num}} \right]$$

*3.4. Model Description*

For our experiments, we used four CNN models (ResNet, ResNeXt, DenseNet, and XceptionNet) that are known to be effective for various image analysis tasks, including medical image classification. The description of each model is as follows.



### 3.4.1. ResNet and ResNeXt

Simonyan et al. [18] confirmed that the deeper the network model, the better the performance through the VGG network. There is a limitation in that the performance does not improve after VGG's layer 16, but ResNet allows the performance to improve even in networks deeper than VGG. When applying Resnet, the deeper network has a lower error rate. Chollet et al. [14] explained that ResNeXt has a simpler structure compared to other models (ResNet-101/152, ResNet200) but has better performance for image analysis in general. Increasing the cardinality results in fewer errors than making ResNet deeper and wider. The detailed configuration of the ResNet and ResNeXt is presented in Table 7, and it is expressed based on the reference.

**Table 7.** Detailed configuration of the ResNet and ResNeXt.

| Stage | Resolution | ResNet-50 Operator | ResNeXt-50 Operator |
|---|---|---|---|
| 1 | $112 \times 112$ | $Conv7 \times 7$ | $Conv7 \times 7$ |
| 2 | $56 \times 56$ | [$Conv1 \times 1$ 64, $Conv3 \times 3$ 64, $Conv1 \times 1$ 256] $\times 3$ | [$Conv1 \times 1$ 128, $Conv3 \times 3$ 128, $Conv1 \times 1$ 256] $\times 3$ |
| 3 | $28 \times 28$ | [$Conv1 \times 1$ 128, $Conv3 \times 3$ 128, $Conv1 \times 1$ 512] $\times 4$ | [$Conv1 \times 1$ 128, $Conv3 \times 3$ 128, $Conv1 \times 1$ 512] $\times 4$ |
| 4 | $14 \times 14$ | [$Conv1 \times 1$ 256, $Conv3 \times 3$ 256, $Conv1 \times 1$ 1024] $\times 6$ | [$Conv1 \times 1$ 512, $Conv3 \times 3$ 512, $Conv1 \times 1$ 1024] $\times 6$ |
| 5 | $7 \times 7$ | [$Conv1 \times 1$ 512, $Conv3 \times 3$ 512, $Conv1 \times 1$ 2048] $\times 3$ | [$Conv1 \times 1$ 1024, $Conv3 \times 3$ 1024, $Conv1 \times 1$ 2048] $\times 3$ |
| 6 | $1 \times 1$ | Average pool $7 \times 7$, FC | Average pool $7 \times 7$, FC |

### 3.4.2. DenseNet

We also used DenseNet because it is a CNN model that is frequently compared to Resnet. Xie et al. [19] described that there are advantages in that the number of parameters can be considerably reduced, feature propagation can be enhanced, and vanishing gradient can be prevented. It performs well and has low computational complexity on representative datasets, such as SVHN and ImageNet. The detailed configuration of the model is given in Table 8.

**Table 8.** Detailed configuration of the DenseNet.

| Stage | Operator | Resolution | #Layers |
|---|---|---|---|
| 1 | $Conv7 \times 7$ | $112 \times 112$ | Convolution |
| 2 | Max pool $3 \times 3$ | $56 \times 56$ | Pooling |
| 3 | [$Conv1 \times 1$, $Conv3 \times 3$] $\times 6$ | $56 \times 56$ | Dense Block |
| 4 | Average pool $2 \times 2$ | $28 \times 28$ | Transition Layer |
| 5 | [$Conv1 \times 1$, $Conv3 \times 3$] $\times 12$ | $28 \times 28$ | Dense Block |
| 6 | Average pool $2 \times 2$ | $14 \times 14$ | Transition Layer |
| 7 | [$Conv1 \times 1$, $Conv3 \times 3$] $\times 24$ | $14 \times 14$ | Dense Block |
| 8 | Average pool $2 \times 2$ | $7 \times 7$ | Transition Layer |
| 9 | [$Conv1 \times 1$, $Conv3 \times 3$] $\times 16$ | $7 \times 7$ | Dense Block |
| 10 | Average pool $7 \times 7$, FC, SoftMax | $1 \times 1$ | Classification Layer |

### 3.4.3. XceptionNet

Chollet et al. [20] explained that XceptionNet is an Extreme Inception model that is known to perform better using parameters of the same capacity. It is a model that adds depth-wise separable convolution to the Inception model. Furthermore, it is a model with ResNet-based separable convolutions trained upon the ImageNet dataset and is known to perform best for various image qualities. A detailed configuration of the model is given in Figure 3. We used XceptionNet using this configuration, since it performed well with a F1 score of 73.19% without augmentation.

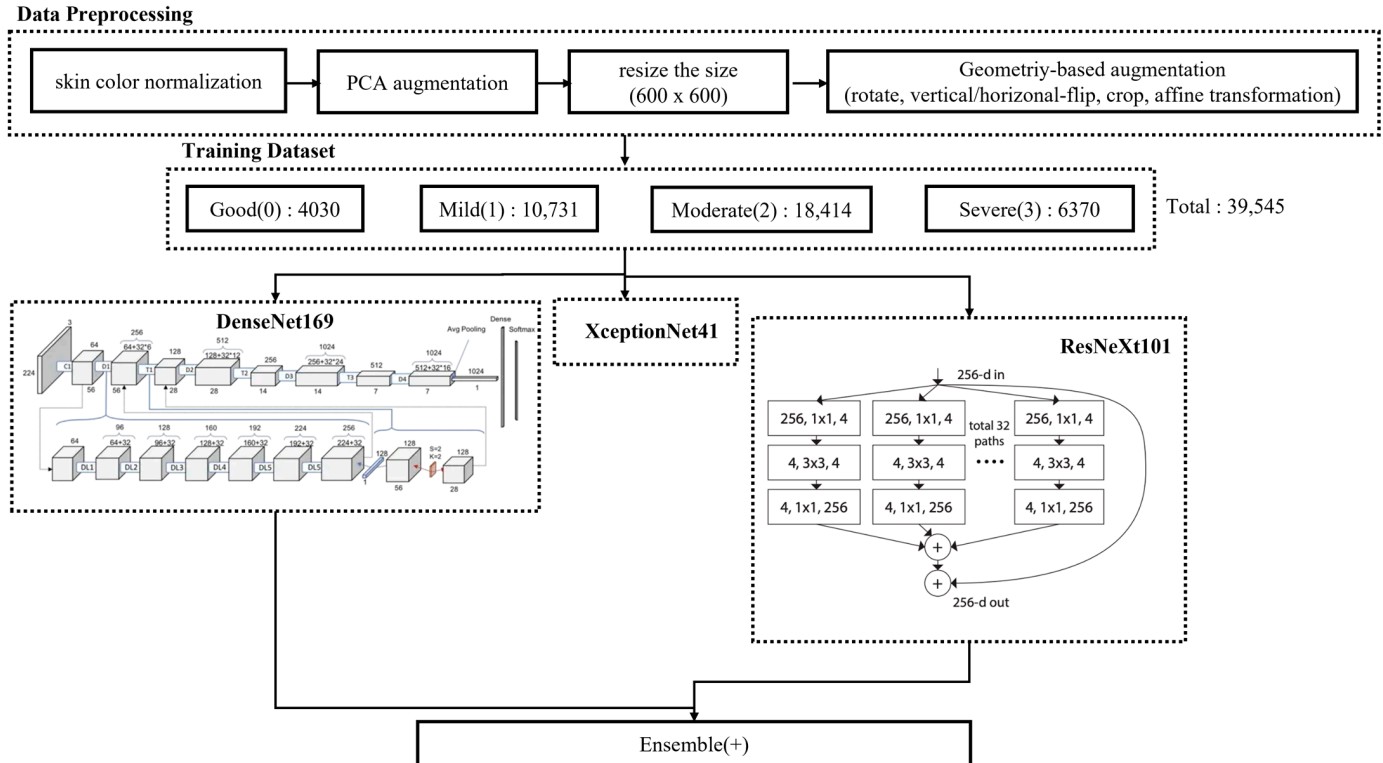

**Figure 3.** Deep learning model architecture for alopecia classification.

The final process for the input data and data augmentation for forming the ensemble is represented in Figure 3.

## 4. Experiment and Result

### 4.1. Experimental Environments

For comparison, we used a model without data augmentation and a model with data augmentation. All experiments were conducted using an Adam optimizer, stepLR scheduler, and the cross-entropy loss function described in Section 3.3, with a batch size of 32 and 30 epochs.

### 4.2. Experimental Result

Table 9 summarizes the best accuracy of each model and the result of learning a model to which the data augmentation process has not been added. The row for each network means the number of layers in network. For example, ResNet101 has 101 layers and ResNet152 has 152 layers. The F1 score and accuracy are presented. Accuracy is the ratio of correctly predicting true/false among all classification results, and the F1 score is a harmonic mean, considering precision and recall value. In this experiment, given that the number of classes is unbalanced, the imbalance of classes was corrected using the F1 score. The F1 score and accuracy are calculated as follows.

The F1 score varies across various classes (scalp conditions). For example, the F1 score for mild (1) is 93.36%; however, the score for moderate (2) is only 21.25%. We believe that such variances are due to the differences in the number of available data for each class. Across the models, DenseNet201 has the highest average F1 score (77.12%) compared to ResNet, ResNeXt, or XceptionNet.

Table 10 lists the F1 scores with data augmentation. Data augmentation was applied to training as well as test datasets, using the augmentation approach is described in Section 3. ResNeXt101 has the highest average F1 score (86.8%) compared to the other models. We noticed that the difference in F1 score for each label also decreased with the data augmentation.

Figure 4 presents the F1 score and accuracy with ensembles of two models that make use of data augmentation. The ensemble model with ResNeXt101 and DenseNet169 has the highest F1 score (87.5%). The ensemble model with DensetNet169 and XceptionNet41 has the highest accuracy (95.84%).

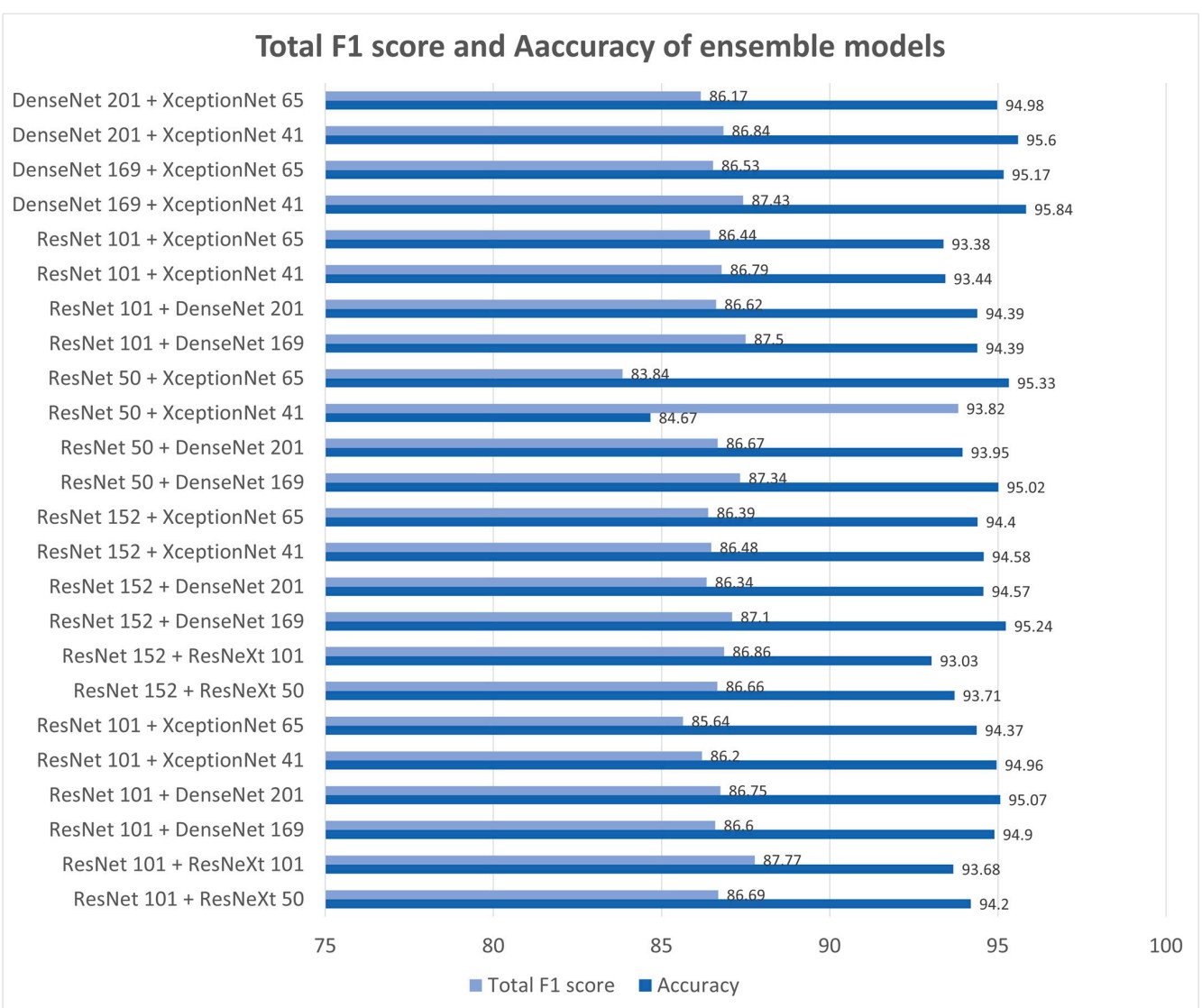

**Figure 4.** F1 score and accuracy of the best ensemble models with data augmentation.

**Table 9.** F1 score of a single network model without data augmentation.

| Network | | Good (0) F1 Score | Mild (1) F1 Score | Moderate (2) F1 Score | Severe (3) F1 Score | Total F1 Score |
|---|---|---|---|---|---|---|
| ResNet | 101 | 83.74 | 83.63 | 49.03 | 55.32 | 75.24 |
| | 152 | 66.67 | 80.41 | 56.76 | 60.64 | 74.16 |
| ResNeXt | 50 | 91.06 | 79.42 | 48.89 | 62.77 | 72.90 |
| | 101 | 89.43 | 78.19 | 56.17 | 63.30 | 73.42 |
| DenseNet | 169 | 89.43 | 86.97 | 38.19 | 65.96 | 76.27 |
| | 201 | 69.11 | 93.36 | 21.25 | 72.87 | 77.12 |
| XceptionNet | 41 | 84.55 | 80.70 | 42.79 | 77.66 | 73.19 |
| | 65 | 76.42 | 80.08 | 49.33 | 63.30 | 72.95 |

Figure 5 lists the F1 score and accuracy of the three models' ensembles with data augmentation. Overall, the performance is further improved. The ensemble model with ResNext101, DenseNet169, and XceptionNet41 has the highest F1 score (87.74%). The ensemble model with DensetNet169, XeptionNet41, and ResNet101 has the highest accuracy (95.75%). The F1 score was 0.24% higher when three models were ensembled, while the accuracy was 0.09% higher when two were ensembled. Furthermore, the F1 score for each label increased more evenly when we ensembled three models.

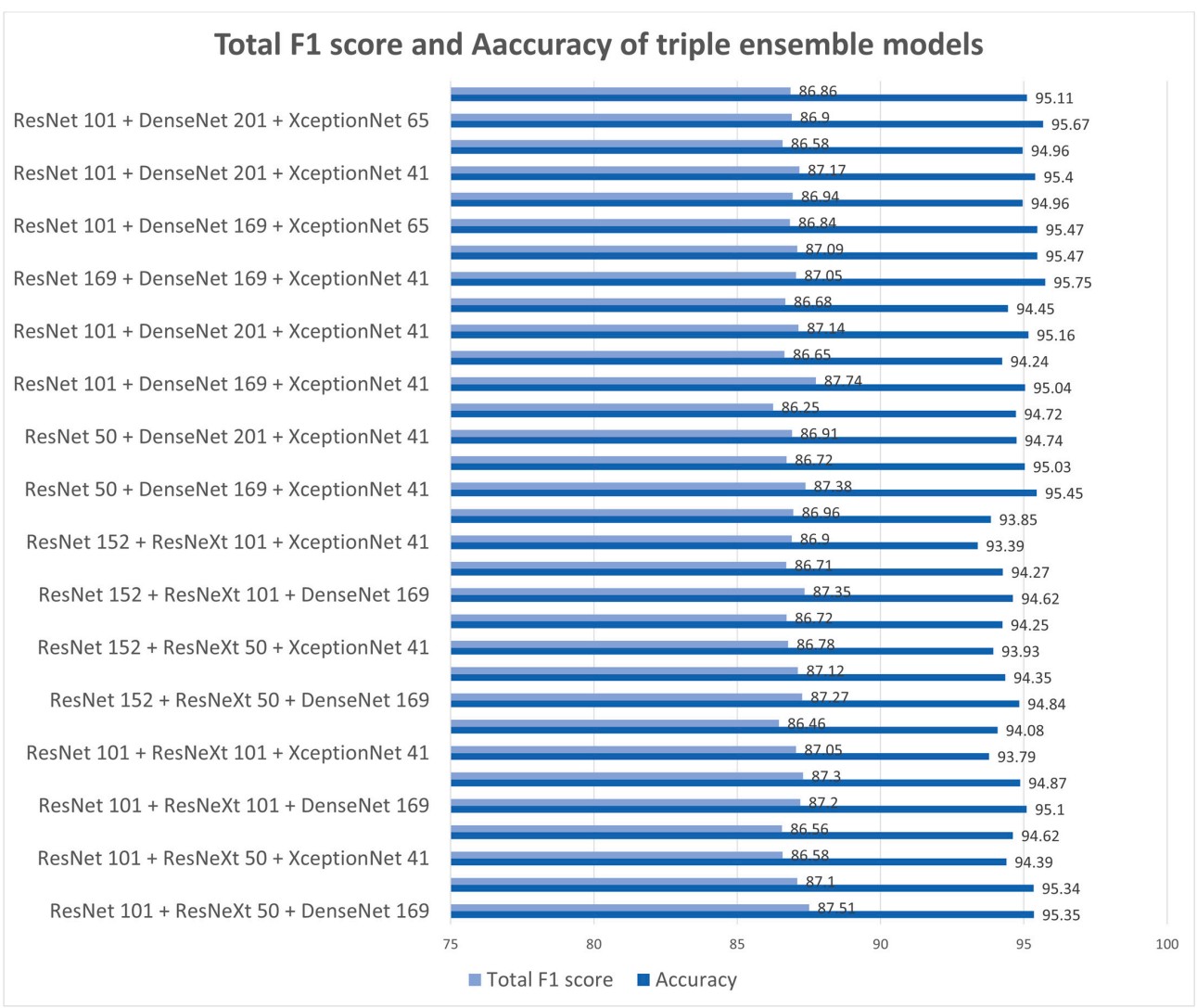

**Figure 5.** F1 score and accuracy of the triple ensemble model with data augmentation.

**Table 10.** F1 score of a single network model with data augmentation.

| Network | | Good (0) F1 Score | Mild (1) F1 Score | Moderate (2) F1 Score | Severe (3) F1 Score | Total F1 Score |
|---|---|---|---|---|---|---|
| ResNet | 101 | 92.68 | 100.0 | 87.87 | 60.43 | 86.17 |
| | 152 | 90.89 | 100.0 | 85.79 | 67.45 | 86.44 |
| ResNeXt | 50 | 90.33 | 100.0 | 91.60 | 56.91 | 86.75 |
| | 101 | 86.91 | 100.0 | 89.55 | 63.78 | 86.80 |
| DenseNet | 169 | 94.63 | 99.79 | 89.10 | 59.15 | 86.64 |
| | 201 | 93.50 | 100.0 | 89.10 | 57.23 | 86.17 |
| XceptionNet | 41 | 90.41 | 99.63 | 90.17 | 54.15 | 85.52 |
| | 65 | 89.11 | 100.0 | 91.43 | 51.22 | 85.41 |

## 5. Conclusions

An increasing number of people suffer from alopecia every year, and it is very difficult to diagnose alopecia early on. It is important to create an AI model that can diagnose alopecia early. However, proper alopecia datasets for the training are not easily found because the data collection requires domain expertise, and the number of available datasets is not very large. This paper presents an approach for improving the classification performance using a set of data augmentation techniques appropriate for scalp images and model ensembles, achieving an accuracy of 95.84% and an F1 score of 87.74%.

Specifically, the number of images was increased by applying geometry-based augmentation through operations such as rotate, vertical flip, horizontal flip, crop, and affine transformation. As the color of the scalp may vary between races, we performed normalization using PCA augmentation, a color-based augmentation technique. For an unbalanced dataset, we applied the focal loss function.

When we evaluated individual classes without data augmentation, the highest F1 score was achieved using DenseNet. When we ensembled two models, ResNext101 and DenseNet169 had the highest F1 score (87.5%). The ensemble model having ResNext101, DenseNet169, and XceptionNet41 achieved the highest F1 score (87.74%). In general, the F1 score for each label increased more evenly when three models were ensembled.

This study used only a microscope image dataset, but for future work, we plan to make use of images captured using regular cameras, which can be used for more general applications.

**Author Contributions:** Conceptualization, M.K., Y.G., Y.K. and J.K.; methodology, M.K., Y.G., Y.K. and J.K.; software, M.K., Y.G. and Y.K.; validation, M.K. and Y.G.; formal analysis, M.K. and Y.G.; investigation, M.K., Y.G. and Y.K.; resources, M.K. and Y.G.; data curation, M.K., Y.G. and Y.K.; writing—original draft preparation, M.K., Y.G. and Y.K.; writing—review and editing, M.K., Y.G., Y.K. and J.K.; visualization, M.K. and Y.G.; supervision, J.K.; project administration, M.K., Y.G. and J.K. All authors have read and agreed to the published version of the manuscript.

**Funding:** This research was supported by the MSIT (Ministry of Science, ICT), Korea, under the High-Potential Individuals Global Training Program (RS-2022-00155054) (50%) and under the ITRC (Information Technology Research Center) support program (IITP-2023-2020-0-01789) (50%) supervised by the IITP (Institute for Information and Communications Technology Planning and Evaluation).

**Institutional Review Board Statement:** Not applicable.

**Informed Consent Statement:** Not applicable.

**Data Availability Statement:** The dataset used in this study is an open access available from AI-Hub: Scalp image by type (https://aihub.or.kr/aihubdata/data/view.do?currMenu=115&topMenu=100&aihubDataSe=realm&dataSetSn=216, accessed on 8 March 2023).

**Conflicts of Interest:** The authors declare no conflict of interest.

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
