# Peer review of "Deep-Learning-Based Scalp Image Analysis Using Limited Data"

_electronics, doi:10.3390/electronics12061380_

Round 1

Reviewer 1 Report

1. Problem statements and research gaps that motivate current study are not evident. Please elaborate.

2. Limited literature survey is covered, particularly in Section 2.2 Existing models for scalp datasets. Please include more related works.

3. More critical literature analysis is needed. For instance, a table can be presented to summarize the existing works. 

4. Table 1 - The images in certain classes are not obviously different. Who is responsible to do the annotation and how to ensure the data annotation is done correctly?

5. Lines 130 - What are the techniques used to remove the shadow and noise appearing at the edge of the original images? Please elaborate.

6. Eq. (5) - Please check if "raito" is a typo.

7. The meaning of some symbols in equations such as p1, p2, p3, alpha(1,2,3), lamba (1,2,3) are not defined. Please ensure all symbols used are properly defined.

8.  Different colors should be used to indicate different results (F1 score and accuracy) in Figures 4 and 5. The current presentation style are quite confusing.

Author Response

Reviewer 1:
1. Problem statements and research gaps that motivate current study are not evident. Please elaborate.
=> Both [3] and [4] use microscope images captured under a portable camera that can be detached from a smartphone. We add this comment in lines 40.

2. Limited literature survey is covered, particularly in Section 2.2 Existing models for scalp datasets. Please include more related works.
=> We add new reference [6] in 2.1 and 2.2.

3. More critical literature analysis is needed. For instance, a table can be presented to summarize the existing works. 
=> We added table to summarize table in part 2.1 and 2.2.

4. Table 1 - The images in certain classes are not obviously different. Who is responsible to do the annotation and how to ensure the data annotation is done correctly?
=> According to the description of the dataset, the classification of scalp state images is a new area with no criteria set globally at the time. Therefore, it is said that three dermatologists at Seoul National University Bundang Hospital set the classification criteria and examined them.

5. Lines 130 - What are the techniques used to remove the shadow and noise appearing at the edge of the original images? Please elaborate.
=> To remove these noises, each image is cropped to a square of 600 pixels based on the center of the data. The shadows that are not removed by cropping process are eliminated through a color conversion. We added this sentence in line 151.

6. Eq. (5) - Please check if "raito" is a typo.
=> "raito" is not a typo, but a range of arbitrarily determined values. In this study, -20% to 20% of the pre-treatment process was finally selected.

7. The meaning of some symbols in equations such as p1, p2, p3, alpha(1,2,3), lamba (1,2,3) are not defined. Please ensure all symbols used are properly defined.
=> We corrected the typos in Equation (6) and Table 4, which incorrectly denoted I_pca as I_p.
Alpha 1,2 and 3 represents the variance and their values are the same. Therefore, all of them are modified as "alpha" to reduce the confusion. We added an explanation that the p_i and λi value are calculated by I_xy.

8.  Different colors should be used to indicate different results (F1 score and accuracy) in Figures 4 and 5. The current presentation style are quite confusing.
=> We edited Figure 4 and 5.

Reviewer 2 Report

The topic presented in the manuscript is interesting. Some aspects should be improved before publication. Please refer to my detailed comments below.

  1. Section 2: the literature needs to be enriched. More recent research should be included and compared.
  2. Figure 1, the number of original dataset is not consistent with the text (line 115 and line 121). Please specify the difference.
  3. The authors should specify how the detailed configurations (Table 5 and Table 6) were selected. Were they taken directly from references, or decided based on analysis?
  4. Similarly, in Figure 3, the authors should comment why this specific configuration was selected.
  5. Table 7 and Table 8, please explain the difference between the two rows for each network, e.g., what do 101 and 152 stand for ResNet?
  6. Figure 4 and Figure 5, for better readability, please indicate which bar corresponds to F1 score and which to accuracy, respectively.
  7. The model architecture (Figure 6) should be presented before demonstrating the results.
  8. The authors should compare the performance of the proposed model with those from existed work to show novelty and advantage.

Author Response

Reviewer 2:
1. Section 2: the literature needs to be enriched. More recent research should be included and compared.
=> We add new reference [6] in 2.1 and 2.2.

2. Figure 1, the number of original dataset is not consistent with the text (line 115 and line 121). Please specify the difference.
=> "526 Good (0), 13,156 Mild (1), 3,742 Moderate (2), and 825 Severe (3)" in line 155 are the number of samples for each class in the original dataset. "18,249" in line 121 is the total amount of original dataset in line 155. And 49,118 is the amount of the final dataset after data augmentation. We added this sentence in 3.1.

3. The authors should specify how the detailed configurations (Table 5 and Table 6) were selected. Were they taken directly from references, or decided based on analysis?
=> The configurations in table 5,6 are taken from [14]. 

4. Similarly, in Figure 3, the authors should comment why this specific configuration was selected.
Table 7 and Table 8, please explain the difference between the two rows for each network, e.g., what do 101 and 152 stand for ResNet?
=> We added an explanation for why this specific configuration was selected in part 3.4.3

5. Figure 4 and Figure 5, for better readability, please indicate which bar corresponds to F1 score and which to accuracy, respectively.
=> Figures 5 and 6 have been modified so that the colors of F1 score and Accuracy are different, and an explanation of each color is added.

6. The model architecture (Figure 6) should be presented before demonstrating the results.
The authors should compare the performance of the proposed model with those from existed work to show novelty and advantage.
=> We moved the model architecture(Figure 6) and its description to section 3.4.

Round 2

Reviewer 1 Report

All comments given have been addressed by authors

Reviewer 2 Report

The reviewer appreciates the authors' consideration of the reviewer's comments. The manuscript is sufficiently modified to the satisfaction of the reviewer.